# Bioinformatic Prediction of Signaling Pathways for Apurinic/Apyrimidinic Endodeoxyribonuclease 1 (APEX1) and Its Role in Cholangiocarcinoma Cells

**DOI:** 10.3390/molecules26092587

**Published:** 2021-04-29

**Authors:** Doungdean Tummanatsakun, Tanakorn Proungvitaya, Sittiruk Roytrakul, Siriporn Proungvitaya

**Affiliations:** 1Centre of Research and Development of Medical Diagnostic Laboratories (CMDL), Faculty of Associated Medical Sciences, KhonKaen University, KhonKaen 40002, Thailand; pui_ddlab41@hotmail.com (D.T.); tanakorn@kku.ac.th (T.P.); 2National Center for Genetic Engineering and Biotechnology (BIOTEC), Functional Ingredients and Food Innovation Research Group, National Science and Technology Development Agency (NSTDA), Pathumthani 12120, Thailand; sittiruk@biotec.or.th; 3Cholangiocarcinoma Research Institute (CARI), Faculty of Medicine, KhonKaen University, KhonKaen 40002, Thailand

**Keywords:** APEX1, CCA, signaling pathway, mass spectrometry, bioinformatics

## Abstract

Apurinic/apyrimidinic endodeoxyribonuclease 1 (APEX1) is involved in the DNA damage repair pathways and associates with the metastasis of several human cancers. However, the signaling pathway of APEX1 in cholangiocarcinoma (CCA) has never been reported. In this study, to predict the signaling pathways of APEX1 and related proteins and their functions, the effects of APEX1 gene silencing on APEX1 and related protein expression in CCA cell lines were investigated using mass spectrometry and bioinformatics tools. Bioinformatic analyses predicted that APEX1 might interact with cell division cycle 42 (CDC42) and son of sevenless homolog 1 (SOS1), which are involved in tumor metastasis. RNA and protein expression levels of APEX1 and its related proteins, retrieved from the Gene Expression Profiling Interactive Analysis (GEPIA) and the Human Protein Atlas databases, revealed that their expressions were higher in CCA than in the normal group. Moreover, higher levels of APEX1 expression and its related proteins were correlated with shorter survival time. In conclusion, the signaling pathway of APEX1 in metastasis might be mediated via CDC42 and SOS1. Furthermore, expression of APEX1 and related proteins is able to predict poor survival of CCA patients.

## 1. Introduction

Cholangiocarcinoma (CCA) is a bile duct cancer originating from biliary epithelial cells [1]. The incidence of CCA is high in China, Korea, and Thailand, particularly in the northeastern region of Thailand, where the incidence of CCA is the highest in the world [2]. In northeastern Thailand, a strong association was observed between CCA and age, *Opisthorchis viverrini* infection, the custom of eating raw cyprinoid fish contaminated with *O**. viverrini* larvae, family history of cancer, and liquor consumption [3]. Surgical resection is the only possible curative treatment for CCA patients [4]. However, recurrence occurs frequently, and the prognosis after resection is not quite satisfactory [4,5]. Therefore, useful biomarkers are necessary, not only for early diagnosis, but also for predicting the prognosis of CCA patients [5]. Although carbohydrate antigen 19-9 (CA19-9) and carcinoembryonic antigen (CEA) have been routinely used as biomarkers for the diagnosis/prognosis of CCA, the sensitivity and specificity of these markers could be improved [5,6,7].

Several studies have focused on the development of prognostic biomarkers for CCA patients. In our previous study, we showed that apurinic/apyrimidinic endodeoxyribonuclease 1 (APEX1) can be a new CCA prognostic marker protein [8], since the serum APEX1 level of CCA patients with lymph node metastasis was significantly higher than that of patients without metastasis [8]. Furthermore, a higher serum APEX1 level was correlated with shorter survival time [8]. Moreover, we demonstrated, using CCA cell lines, that APEX1 plays a role in cell migration and invasion [8]. APEX1 is a multifunctional protein, and its endonuclease activity is involved in the DNA base repair pathway and modulates the redox status of several types of transcriptional factor [9,10,11]. Involvement of APEX1 in tumor metastatic mechanisms has been shown in many cancers [12,13,14], but its role and mechanism in CCA remains unclear. In this study, therefore, using a combination of gene-silencing, mass spectrometry technique, and bioinformatics, we investigated the possible signal transduction pathways of APEX1 in CCA cells. The results presented here showed that APEX1 protein is involved in the metastasis of CCA via CDC42 and SOS1. Expression of APEX1 and its related proteins in CCA patients was higher than in the normal group. Moreover, higher levels of APEX1 expression and its related proteins were correlated with shorter survival time. Thus, APEX1 and its related proteins could be potential markers for poor prognosis of CCA.

## 2. Results

### 2.1. Expression of APEX1 in Cell Lines

To analyze the signaling pathway of APEX1 in the metastasis of CCA, we examined the APEX1 expression in three CCA cell lines, KKU-213A, KKU-213B, and KKU-100, and an immortalized cholangiocyte MMNK1, using western blot analysis. The APEX1 expression of the three CCA cell lines was higher than that of MMNK1 (Figure 1A,B), with the highest expression in KKU-213A and the lowest in KKU-100. Thus, those two cell lines were used as representative cells for gene silencing experiments.

### 2.2. Silencing of APEX1 in CCA Cell Lines

To investigate the possible APEX1-related signaling pathway in CCA cell lines, APEX1 gene of KKU-213A and KKU-100 was silenced using siRNA. The expression of APEX1 in KKU-213A and KKU-100 was successfully suppressed at 24 h after silencing compared with the siRNA scramble control (Figure 2A,B). Thus, the protein expression patterns of APEX1-silenced and scramble-treated KKU-213A and KKU-100 were examined using mass spectrometry.

### 2.3. Protein Expression Patterns of APEX1-Gene-Silenced and Scramble-Treated KKU-213A and KKU-100 Cell Lines

As the first step of protein identification and prediction of the APEX1-related signaling pathway of CCA, protein expression patterns of the cell lysates of APEX1 gene-silenced and scramble-treated KKU-213A and KKU-100 cell lines were analyzed using mass spectrometry followed by Venn diagram, using jvenn software. As shown in Figure 3, we identified 1149 proteins from scramble-treated KKU-213A, 961 from siRNA-treated KKU-213A, 962 from scramble-treated KKU-100 and 1015 from siRNA-treated KKU-100. To identify the proteins related to APEX1 function, 229 proteins that were commonly expressed in scramble KKU-213A and KKU-100 cells but not expressed in siRNA treated cells were selected (Appendix A).

### 2.4. Prediction of APEX1-Related Signaling Pathway

Since 229 proteins were commonly expressed only in scramble-treated cell lines but not in siRNA-treated cells, we selected them for further identification of potential proteins related to APEX1 function. The APEX1-related signaling pathway was predicted using STITCH version 5.0. The interactions between APEX1 and related proteins showed many possible signaling pathways (Appendix A). Among them, we focused only on APEX1-related proteins involved in tumor metastasis. Interestingly, only CDC42 and SOS1 were suppressed after being knocked down by siRNA-APEX1. Moreover, APEX1 might interact with CDC42 and SOS1 via hypoxia-inducible factor-1alpha (HIF-1α) and vascular endothelial growth factor-A (VEGFA). Moreover, the interaction of CDC42 and SOS1 is related to RAF proto-oncogene serine/threonine-protein kinase (RAF1) and mitogen-activated protein kinase 1 (MAPK1) (Figure 4).

### 2.5. Molecular Docking of APEX1 and Related Proteins

To assess binding predictions of APEX1 and the related interacting-proteins predicted by STITCH, the binding geometries of corresponding protein–protein interactions were explored using AutoDockTools for docking analysis. According to the proteins selected by STITCH, we ranked the docking between APEX1 and HIF-1α, VEGFA and HIF-1α, CDC42 and VEGFA, and SOS1 and VEGFA. We searched the three-dimensional (3D) X-ray crystal structure of human APEX1 (PDB ID: 4QHE, solution 1.4 Å), HIF-1α (PDB ID: 4H6J with solution 1.52 Å), VEGFA (PDB ID: 3QTK with solution 1.849 Å), CDC42 (PDB ID: 2QRZ with solution 2.4 Å), and SOS1 (PDB ID: 5OVE with solution 1.85 Å) from the Protein Data Bank. The cutoff for reliability was considered based on the docking conformer of the ligands with a reference root mean square deviation (RMSD) <2 Å. After docking analysis, the RMSD between APEX1 and HIF-1α was 0.53 Å (Figure 5A), between HIF-1α and VEGFA it was 0.53 Å (Figure 5B), between CDC42 and VEGFA it was 0.71 Å (Figure 5C), and between SOS1 and VEGFA it was 0.46 Å (Figure 5D). The results of protein–protein interaction from the docking analysis were in accordance with the prediction by STITCH. Thus, APEX1 might interact with CDC42 and SOS1 via hypoxia-inducible factor-1alpha (HIF-1α) and vascular endothelial growth factor-A (VEGFA).

### 2.6. RNA Expression Levels of APEX1 and Related Protein Genes

Since expression of CDC42 and SOS1 molecules were suppressed after being APEX1 gene knocked down by siRNA, they were considered as APEX1-related proteins (Figure 4). Then, we analyzed the gene expression levels of APEX1 and related proteins at an RNA level between CCA and normal tissues, using the GEPIA database [15]. The results demonstrated that APEX1 RNA was significantly over-expressed in CCA compared with normal control tissue samples (Figure 6A). Likewise, the RNA level of CDC42 in CCA tissues was significantly higher than that of the normal tissue samples (Figure 6B). Moreover, the RNA level of SOS1 in CCA tissues was significantly higher than that of normal tissue samples (Figure 6C).

### 2.7. Protein Expression Levels of APEX1 and Related Proteins in CCA Tissues

To elucidate whether APEX1 and related proteins play a role in cancer progression, we evaluated the protein expression levels of APEX1 and related proteins. Immunohistochemical staining data of APEX1 (9 normal and 16 cancerous tissues), CDC42 (6 normal and 10 cancerous tissues), and SOS1 (3 normal and 5 cancerous tissues) in normal and CCA tissues were retrieved from the Human Protein Atlas database [16]. The correlation between protein expression levels and the patients’ survival time was also analyzed using the Human Protein Database [16]. The results show that APEX1 (Figure 7A,B), CDC42 (Figure 7C,D), and SOS1 staining (Figure 7E,F) were higher in CCA than in normal tissues. The database analysis also provided that the overall survival time of CCA patients with a high APEX1 level (Log Rank *p* < 0.0001; Figure 7G), and also with a high CDC42 level (Log Rank *p* < 0.0001; Figure 7H), was significantly shorter than those of having lower APEX1 or CDC42 levels. Moreover, the overall survival time of CCA patients with a high SOS1 level was significantly shorter than that of CCA patients with a low SOS1 level (Log Rank *p* = 0.0017; Figure 7I).

## 3. Discussion

In this study, we have shown that the expression of APEX1 in three CCA cell lines including KKU-213A, KKU-213B, and KKU-100 was higher than that in an immortalized cholangiocyte cell line named MMNK1. Moreover, the RNA expression levels of APEX1, CDC42, and SOS1 in CCA patients obtained from the GEPIA database were higher than those in the normal group [15]. Protein expression levels of APEX1, CDC42, and SOS1 in CCA patients retrieved from the Human Protein database [16] were higher than in the normal group. Moreover, the Human Protein database (HPD) analysis revealed that CCA patients with a higher expression level of APEX1, CDC42, and SOS1 had a shorter survival time [16]. Thus, APEX1, CDC42, and SOS1 might be associated with CCA progression.

In our previous report, APEX1 knockdown of CCA cell lines resulted in suppression of migration and invasion [8]. Interactions of APEX1 and the metastasis mechanism of related proteins have been reported in several malignant tumors. For example, APEX1 promotes angiogenesis of osteosarcoma through transforming growth factor β (TGF-β) [17]. In addition, APEX1 regulates migration and invasion through epithelial–mesenchymal transition (EMT) in non-small cell lung cancer [13]. Wang et al. (2007) reported that APEX1 regulated vascular endothelial growth factor (VEGF) and invasion through hypoxia-inducible factor-1α (HIF-1α) in osteosarcoma [18]. Jou et al. (2010) demonstrated that CDC42 induced migration and invasion of hepatocellular carcinoma (HCC) [19]. Reymond et al. (2012) found that transient CDC42 depletion decreased endothelial attachment through interaction with beta 1 integrin in lung cancer, suggesting its important role for metastasis [20]. Furthermore, inhibitor of CDC42 can suppress cell cycle progression, proliferation, and migration in lung cancer and prostate cancer [21]. Hillig et al. (2019) suggested that SOS1 inhibition resulted in a reduction of extracellular signal-regulated kinase (ERK), which was involved in the progression of lung cancer [22]. Timofeeva et al. (2009) reported that the depletion of SOS1 in prostate cancer resulted in decreased capacities for cell proliferation, migration, and invasion through inhibition of ERK1/2 [23]. Shinde et al. (2019) suggested that assessment of spleen tyrosine kinase activity is a biomarker for metastasis in breast cancer [24]. Moreover, the same research group reported that epithelial–mesenchymal plasticity contributes to metastatic niche development and distant metastasis of breast cancer. This process is controlled via tumor-derived extracellular vesicles, which contain aberrant levels of transglutaminase-2 and fibrillar fibronectin [25]. Thus, APEX1 might be associated with CDC42 and SOS1 and involved in the metastatic process of CCA.

## 4. Materials and Methods

### 4.1. Human Cell Lines

Three CCA cell lines, KKU-213A [26], KKU-213B [26], and KKU-100 [27], were established from CCA patients and obtained from the Japanese Collection of Research Bioresources (JCRB) Cell Bank, Osaka, Japan. MMNK1 was established and characterized by Maruyama et al. [28]. All cell lines were cultured in Ham’s F-12 (Life technologies, Grand Island, NY, USA) supplemented with 10% heat-inactivated fetal bovine serum, 100 U/mL of penicillin, and 100 µg/mL of streptomycin (Life technologies, Grand Island, NY, USA), and incubated with 5% CO_2_ air atmosphere at 37 °C. The cells were subcultured every three or four days. All cell lines in this study were confirmed mycoplasma-free by specific PCR.

### 4.2. Western Blot Analysis

To check APEX1 protein expression, the cells were harvested in RIPA lysis buffer (150 mM NaCl, 50 mM Tris-HCl, 1% (*v*/*v*) Tween-20, 1% (*w*/*v*) sodium deoxycholate, 0.1% (*w*/*v*) SDS, protease, and phosphatase inhibitors cocktail) and incubated at 4 °C for 10 min. The cell lysate was centrifuged at 20,000× *g* at 4 °C for 30 min. The level of APEX1 protein was determined using western blot analysis with β-actin as a loading control. The experiments were done in triplicate.

Thirty micrograms of protein samples of cell lysate were dissolved in SDS loading buffer (10% sodium dodecyl sulfate (SDS), 1M Tris-HCl, pH 6.8), and boiled for 5 min. Protein concentration was determined by Bradford assay. The samples were, in parallel with standard molecular weight markers, separated on SDS-PAGE (4% stacking gel and 12.5% separating gel) at 150 V for 2 h at room temperature (RT). After electrophoresis, proteins were transferred onto a PVDF membrane (GE Healthcare, Buckinghamshire, UK) for 1 h at RT. The membrane was blocked with 5% skim milk in Tris-buffered saline with 0.1% Tween-20 (TBST, pH 7.4) for 1 h at RT. The membrane was then incubated with 1:500 dilution of rabbit polyclonal antibody against human APEX1 (Cat#orb129513, Biorbyt, Cambridge, UK) overnight at 4 °C. The membrane was washed with 1X TBST, incubated with a 1:10,000 dilution of horseradish peroxidase-conjugated goat anti-rabbit IgG secondary antibody for 1 h at RT, and washed with 1X TBST. Finally, peroxidase activity was detected as chemiluminescence using an ECL Prime Western blotting detection reagent (GE Healthcare, Buckinghamshire UK) and quantitatively analyzed using an Amersham Imager 600 (GE Healthcare Bio-Sciences AB, Uppsala, Sweden).

### 4.3. Transient Silencing of APEX1 Gene Using siRNA

Since APEX1 expression was highest in the KKU-213A cell lysate and lowest in KKU-100 cell lysate, we selected these two cell lines as the representatives for gene silencing experiments. For the APEX1 gene silencing using a siRNA technique, the cells (1.5 × 10^5^ cells/well) were seeded in a 6-well plate and cultured overnight before being transfected with 100 pM of siAPEX1 (Cat#orb260731, Biorbyt, Cambridge, UK), while scrambled siRNA (Invitrogen, Carlsbad, CA, USA) was used as a negative control. Transfection was carried out using Lipofectamine 2000 (Invitrogen, Carlsbad, CA, USA) according to the manufacturer’s instructions. After 6 h of transfection, the culture medium was replaced with complete medium, and the plates were incubated at 37 °C for 24 h. APEX1 expression levels were determined using western blot analysis and further analyzed by mass spectrometry technique. The experiments were done in triplicate.

### 4.4. Gel Formation and Tryptic Digestion

After successful APEX1 gene-silencing by transfection of siRNA, 4 µg of cell lysates of APEX1-silenced and scramble control cells were used for the gel formation in a microtube. The gel was cut into small pieces, and the gel pieces were then subjected to in-gel digestion using an in-house method developed by the Functional Proteomics Technology Laboratory, National Center for Genetic Engineering and Biotechnology (BIOTEC), Thailand. Briefly, samples were completely dissolved in 10 mM ammonium bicarbonate (AMBIC), reduced disulfide bonds using 5 mM dithiothreitol (DTT) in 10 mM AMBIC at 60 °C for 1 h, and alkylated sulfhydryl groups using 15 mM iodoacetamide (IAA) in 10 mM AMBIC at RT for 45 min in the dark. For digestion, samples were mixed with 50 ng/µl of sequencing grade trypsin (1:20 ratio) (Promega, Germany) and incubated at 37 °C overnight. Prior to LC-MS/MS analysis, the digested samples were dried and protonated with 0.1% formic acid before injection into LC-MS/MS.

### 4.5. Liquid Chromatography-Tandem Mass Spectrometry (LC/MS-MS)

The tryptic peptide samples were prepared for injection into an Ultimate3000 Nano/Capillary LC System (Thermo Scientific, Altrincham, UK) coupled to a Hybrid quadrupole Q-Tof impact II™ (Bruker Daltonics, Bremen, Germany) equipped with a Nano-captive spray ion source. Briefly, peptides were enriched on a µ-Precolumn 300 µm I.D. × 5 mm C18 Pepmap 100, 5 µm, 100 A (Thermo Scientific, UK), separated on a 75 μm I.D. × 15 cm, and packed with Acclaim PepMap RSLC C18, 2 μm, 100Å, nanoViper (Thermo Scientific, UK). Solvent A and B, containing 0.1% formic acid in water and 0.1% formic acid in 80% acetonitrile, respectively, were supplied on the analytical column. A gradient of 5–55% solvent B was used to elute the peptides at a constant flow rate of 0.30 μL/min for 30 min. Electrospray ionization was carried out at 1.6 kV using the CaptiveSpray. Mass spectra (MS) and MS/MS spectra were obtained in the positive-ion mode over the range (*m*/*z*) 150–2200 (Compass 1.9 software, Bruker Daltonics version 4.4). The LC-MS analysis of each sample was done in triplicate.

### 4.6. Protein Quantitation and Identification

MaxQuant version 1.6.6.0 was used to quantify the proteins in individual samples using the Andromeda search engine to correlate MS/MS spectra to the Uniprot Homo sapiens database [29]. Label-free quantitation with MaxQuant’s standard settings was performed: a maximum of two miss cleavages, mass tolerance of 0.6 dalton for the main search, trypsin as digesting enzyme, carbamidomethylation of cystein as fixed modification, and the oxidation of methionine and acetylation of the protein N-terminus as variable modifications. Only peptides with a minimum of 7 amino acids, as well as at least one unique peptide, were required for protein identification. Only proteins with at least two peptides, and at least one unique peptide, were considered as being identified and used for further data analysis. Protein FDR was set at 1% and estimated by using the reversed search sequences. The maximal number of modifications per peptide was set to 5. As a search FASTA file, the proteins present in the Homo sapiens proteome were downloaded from Uniprot on 27 April 2020. Potential contaminants presented in the contaminants.fasta file that came with MaxQuant were automatically added to the search space by the software.

The MaxQuant ProteinGroups.txt file was loaded into Perseus version 1.6.6.0 [30], and potential contaminants that did not correspond to any UPS1 protein were removed from the data set. Max intensities were log2 transformed and pairwise comparisons between conditions were made via t-tests. Missing values were also imputed in Perseus using a constant value (zero). The final data containing protein name, accession number, peptide sequence, Q-value, ID score, and signal intensity were exported into excel files.

### 4.7. Selection of Candidate Proteins Using Bioinformatics Tool

Efficacy of APEX1 silencing was presented by mass spectrometric data with the signal intensity of all proteins of cell lysates before and after APEX1 silencing of KKU-213A and KKU-100 cell lines. Overlapping and unique proteins were identified using jvenn software (http://jvenn.toulouse.inra.fr/app/example.html/ (Accessed on 11 November 2020) [31]. To speculate on the active role(s) of APEX1 in CCA and its possible signaling pathway, proteins that were uniquely expressed in scramble-treated KKU-213A and KKU-100 cells but not siRNA-treated cells were selected as the candidate protein group.

### 4.8. Prediction of APEX1 Signaling Pathway

The potential interaction of the identified proteins was analyzed using the search tool for interaction of chemicals and proteins, STITCH 5.0 (http://stitch.embl.de/, accessed on 15 November 2020) [32]. In brief, 229 proteins that were expressed only in scramble-treated CCA cells and APEX1 were put into a box of protein names in a multiple name item to predict the APEX1 related signaling pathway. Then “*Homo sapiens*” was selected as the organism, and then we clicked to continue. The page showed a list names of proteins, and then we clicked to continue. The page showed the confidence view. Stronger associations are represented by the thicker lines, weak associations are represented by thin lines and protein–protein interactions are shown in solid lines.

### 4.9. Molecular Docking

The strength of the interactions was quantified by the affinity energy of ligands for protein targets using the open software AutoDockTools [33]. The entire process was done in the Cygwin from the Harvard-MIT Division of Health Sciences and Technology. The docking flow has several steps that included ligand and protein processing, conversion, and geometric optimization before docking calculations [34]. Briefly, the compounds for all proteins were downloaded as PDB 3D models from the Protein Data Bank [35]. The proteins with high-resolution structures (<3 Å) were chosen for molecular docking [35,36], and the non-protein part, e.g., water molecules, other ligands, etc., was eliminated. The PDB 3D models of proteins were converted into PDBQT format using AutoDockTools [33]. The protein target was considered rigid in all docking calculations and the interaction searching considered the entire surface of the targets, which followed the methods of previous research [34]. The docking flow was based on Cygwin, including the reading of the final results. The cutoff for reliability was considered based on the docking conformer of the ligands with the reference root mean square deviation (RMSD) <2 Å [36,37,38].

### 4.10. Evaluation of RNA Expression Levels in Tissue

To evaluate RNA expression levels of APEX1 and related proteins between CCA and normal tissues, data sets were retrieved and analyzed using the search tool, Gene Expression Profiling Interactive Analysis (GEPIA, http://gepia.cancer-pku.cn/, accessed on 2 March 2021) [15]. Gene name was input into a text box, and then we clicked to “Boxplots”. The page showed datasets selection (cancer name), and then we added cancer type and clicked to plot. The page showed the boxplot of RNA expression between cancer and normal tissues.

### 4.11. Protein Expression in Tissue and Survival Analysis

To evaluate protein expression levels of APEX1 and related proteins between CCA and normal tissues, the data were extracted and analyzed using the search tool, the Human Protein atlas (https://www.proteinatlas.org/, accessed on 5 March 2021) [16]. Protein name was input into a text box, and then clicked to “Search”. The page showed a list names of proteins, and then we selected the tissue. The page showed cancer type, and then we selected CCA. The page showed protein expression in CCA and normal tissues by immunohistochemical method. Moreover, the page showed the survival time analysis of CCA.

## 5. Conclusions

In the present study, APEX1 molecule was associated with metastasis in CCA and was possibly shown to be related with CDC42 and SOS1 by STITCH software. AutoDockTools showed that APEX1 can be bound with CDC42 and SOS1. Thus, we speculated that the signaling pathway of APEX1 in metastasis is related to CDC42 and SOS1. Moreover, a higher level of APEX1 expression and its related proteins was correlated with shorter survival time. Therefore, APEX1 and its related proteins were able to be applied in clinical prognosis.

## Figures and Tables

**Figure 1 molecules-26-02587-f001:**
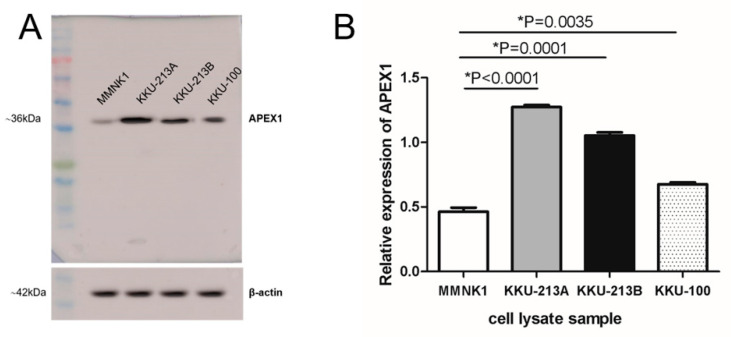
Expression of APEX1 in three CCA cell lines and immortalized cholangiocyte cell line MMNK1. (**A**) Western blot analysis showed a high expression of APEX1 in KKU-213A and KKU-213B. β-actin was used as a control for protein loading. (**B**) APEX1 protein expression was determined using quantitative image analysis. Relative expression of APEX1 was the intensity ratio of APEX1 and β-actin.

**Figure 2 molecules-26-02587-f002:**
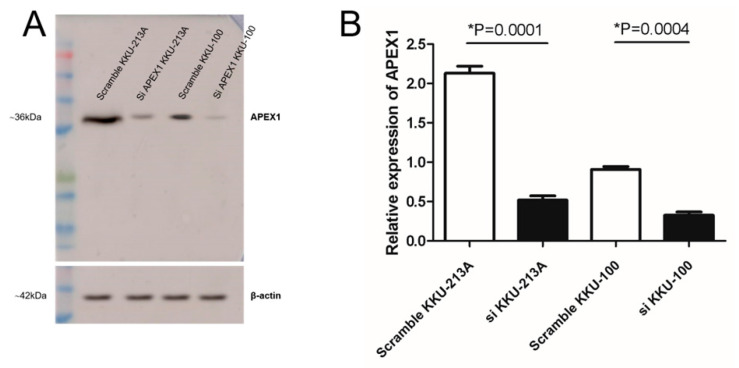
Expression of APEX1 protein in APEX1 gene-silenced, compared with scramble-control, CCA cell lines. (**A**) Western blot analysis showed the suppression of APEX1 protein expression after gene silencing. β-actin was used as a control for loading protein. (**B**) Relative expression of APEX1 for gene silencing.

**Figure 3 molecules-26-02587-f003:**
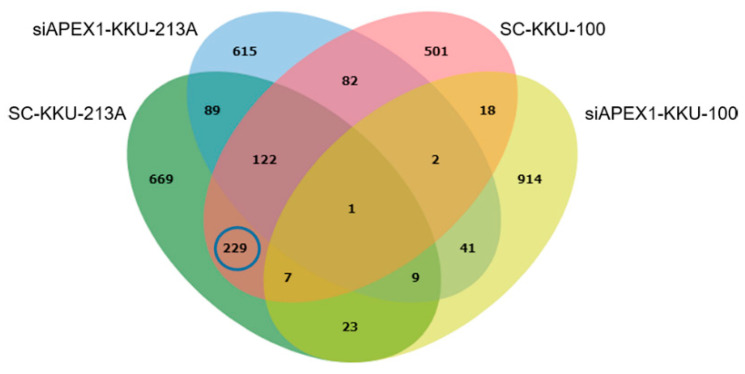
Venn diagram presenting the number of proteins in each sample and the degree of individual overlap among APEX1-gene silenced and scramble-treated KKU-213A and KKU-100 cells.

**Figure 4 molecules-26-02587-f004:**
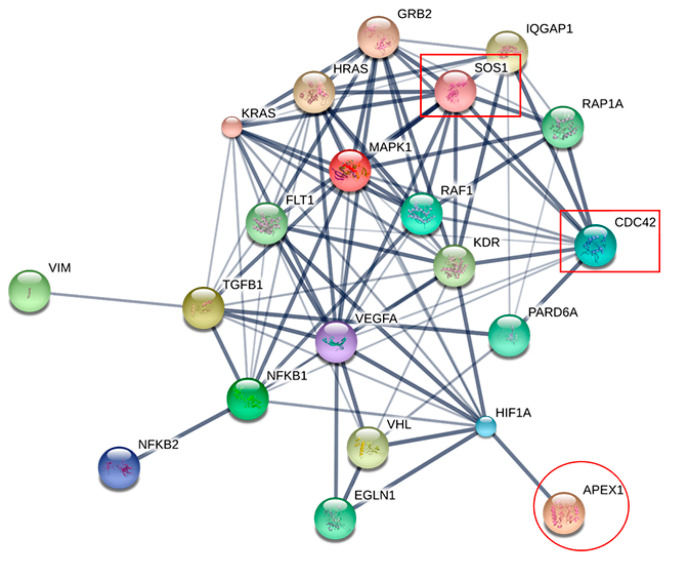
The interaction map of APEX1 and metastatic process-associated proteins. APEX1 was associated with metastatic protein markers: cell division cycle 42 (CDC42) and son of sevenless homolog 1 (SOS1). Stronger associations are represented by thicker lines. Weak associations are represented by thin lines.

**Figure 5 molecules-26-02587-f005:**
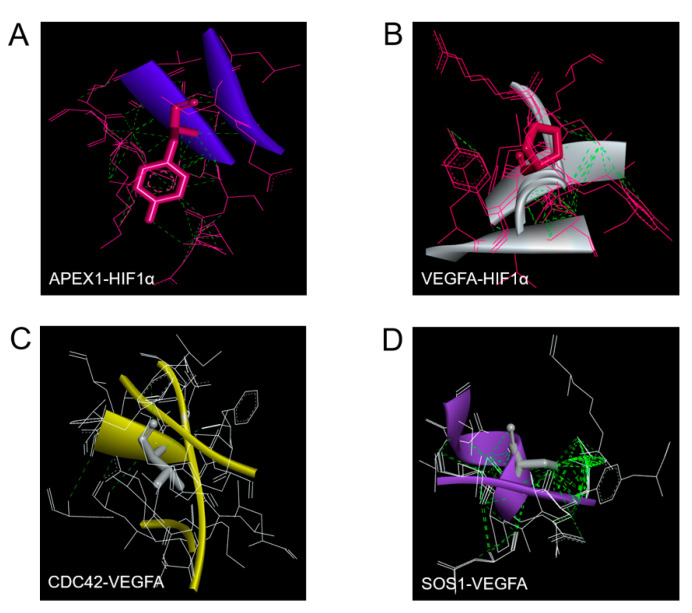
Representative positions of protein–protein docking. Green dash line represents the interaction bond. (**A**) Blue ribbon represents APEX1 interface to HIF-1α (pink). (**B**) White ribbon represents VEGFA interface to HIF-1α (pink). (**C**) Yellow ribbon represents CDC42 interface to VEGFA (white). (**D**) Violet ribbon represents SOS1 interface to VEGFA (white).

**Figure 6 molecules-26-02587-f006:**
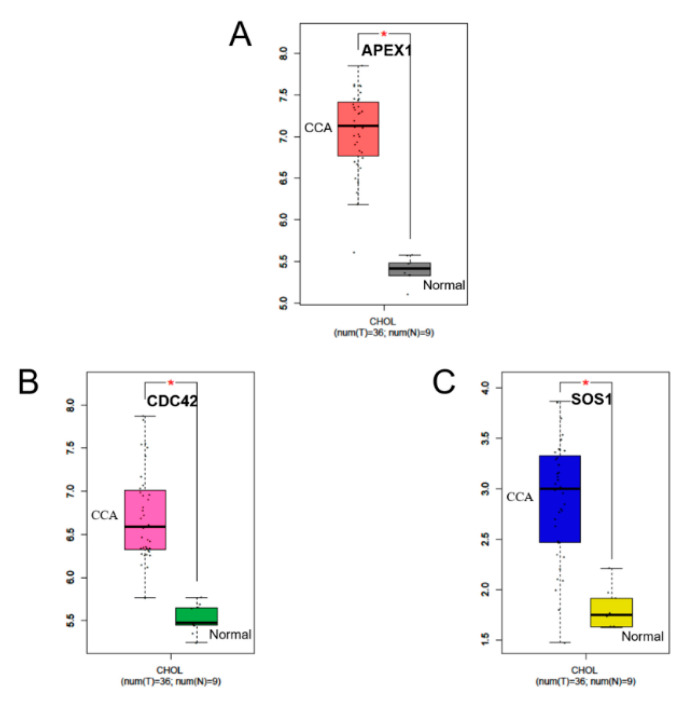
Box diagram of the expression of the RNA level between CCA and normal tissues. Image credit: GEPIA, http://gepia.cancer-pku.cn/ (Accessed on 2 March 2021) [15]. CCA tissue samples are represented by the left box. Normal tissue samples are represented by the right box. RNA expression levels of (**A**) APEX1, (**B**) CDC42, and (**C**) SOS1.

**Figure 7 molecules-26-02587-f007:**
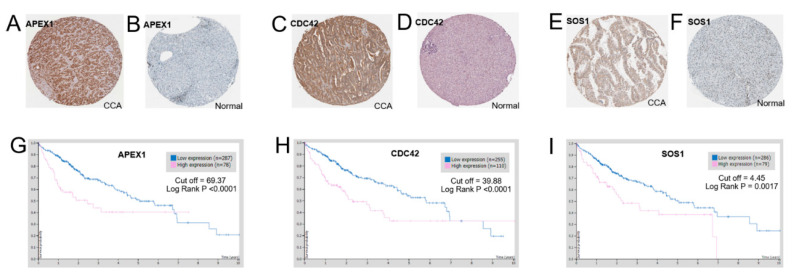
Representative pictures for immunohistochemical detection in CCA and normal tissue, and overall survival time. Image credit: Human Protein Atlas, www.proteinatlas.org, [16]. Image available at the following URL: v20.proteinatlas.org/humancell/ (Accessed on 5 March 2021) (**A**) APEX1 staining of CCA tissue. (**B**) APEX1 staining of normal tissue. (**C**) CDC42 staining of CCA tissue. (**D**) CCD42 staining of normal tissue. (E) SOS1 staining of CCA tissue. (**F**) SOS1 staining of normal tissue. (**G**) APEX1 staining and overall survival time. (**H**) CDC42 staining and overall survival time (**I**) SOS1 staining and overall survival time.

## Data Availability

Not applicable.

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
