# Peer review of "Bioinformatic Prediction of Signaling Pathways for Apurinic/Apyrimidinic Endodeoxyribonuclease 1 (APEX1) and Its Role in Cholangiocarcinoma Cells"

_molecules, 2021, doi:10.3390/molecules26092587_

Round 1
Reviewer 1 Report
Doungdean Tummanatsakun et al in this work analyzed the protein
Apurinic/apyrimidinic endodeoxyribonuclease 1 (APEX1) in the growth of
cholangiocarcinoma (CCA). They combined the shAPEX1 and the bioinformatic analysis and predicted that APEX1 could trigger the signaling of CDC42 and SOS1, thus promote the metastasis and the poor survival of CCA patients. This work lack functional studies, they should analyze the signaling related to the CDC42 and SOS1 in their shAPEX1 cells to support their conclusion. In addition, there is no data support their metastasis aspect of CCA cancer, the title should be changed or the metastasis assay should be performed.
Author Response
We change the tiltle on page 1

Reviewer 2 Report
This is the observational study with moderate Scientific Soundness
Author Response
Thank you for your kind evaluation.

Reviewer 3 Report
The studies are nicely executed, and most of the findings are straight forward. While the concept of the work is very interesting and informative, there are a few concerns that the authors need to address in their study.
- Elaborate more on the significance and big picture of the research in the introduction.
- Please improve discussion section of the manuscript.
- Minor grammatical mistakes needs to be fixed
- Please cite the following articles in your manuscript and include in reference section
- Transglutaminase-2 facilitates extracellular vesicle-mediated establishment of the metastatic niche
- Spleen Tyrosine Kinase–Mediated Autophagy Is Required for Epithelial–Mesenchymal Plasticity and Metastasis in Breast Cancer.
Author Response
- Introduction section in paragraph 2; line 14-18; we have elaborate more on the significance and big picture of the research at the end of the introduction on page 2.
- Discussion section in paragraph 2; line 17-22; we have included two papers of Shinde et al. (2019; 2020) as suggested by the reviewer 3 and expanded the discussion on page 7-8.
- References; we have cited two references suggested in the Discussion section and they were added to the Reference list as No. 36 and 37 on page 14.
-
English usage has been checked by the Publication Clinic staff who is the expert of English editing.
